

# Exploring snake occurrence records: Spatial biases and marginal gains from accessible social media

Benjamin M. Marshall and Colin T. Strine

School of Biology, Institute of Science, Suranaree University of Technology, Nakhon Ratchasima, Nakhon Ratchasima, Thailand

## ABSTRACT

A species' distribution provides fundamental information on: climatic niche, biogeography, and conservation status. Species distribution models often use occurrence records from biodiversity databases, subject to spatial and taxonomic biases. Deficiencies in occurrence data can lead to incomplete species distribution estimates. We can incorporate other data sources to supplement occurrence datasets. The general public is creating (via GPS-enabled cameras to photograph wildlife) incidental occurrence records that may present an opportunity to improve species distribution models. We investigated (1) occurrence data of a cryptic group of animals: non-marine snakes, in a biodiversity database (Global Biodiversity Information Facility (GBIF)) and determined (2) whether incidental occurrence records extracted from geo-tagged social media images (Flickr) could improve distribution models for 18 tropical snake species. We provide R code to search for and extract data from images using Flickr's API. We show the biodiversity database's 302,386 records disproportionately originate from North America, Europe and Oceania (250,063, 82.7%), with substantial gaps in tropical areas that host the highest snake diversity. North America, Europe and Oceania averaged several hundred records per species; whereas Asia, Africa and South America averaged less than 35 per species. Occurrence density showed similar patterns; Asia, Africa and South America have roughly ten-fold fewer records per 100 km$^2$ than other regions. Social media provided 44,687 potential records. However, including them in distribution models only marginally impacted niche estimations; niche overlap indices were consistently over 0.9. Similarly, we show negligible differences in Maxent model performance between models trained using GBIF-only and Flickr-supplemented datasets. Model performance appeared dependent on species, rather than number of occurrences or training dataset. We suggest that for tropical snakes, accessible social media currently fails to deliver appreciable benefits for estimating species distributions; but due to the variation between species and the rapid growth in social media data, may still be worth considering in future contexts.

Corresponding authors
Benjamin M. Marshall,
benjaminmichaelmarshall@gmail.com
Colin T. Strine,
strine.conservation@gmail.com

## INTRODUCTION

Species distribution models can yield insight into a species' niche and habitat (*Santos et al., 2006*). Information on a species' niche provides some ability to predict species'

responses to environmental change (*Penman et al., 2010*; *Yousefi et al., 2015*; *Ahmadi et al., 2019*). Predictions from species distribution models can also inform protected area allocation (*Tulloch et al., 2016*), support conservation status assessments (*Solano & Feria, 2007*; *Fourcade et al., 2013*), invasion potential (*Pearson, 2015*; *Mutascio et al., 2018*; although with complications *Phillips, Chipperfield & Kearney, 2008*) and identify potential human-wildlife conflicts (*Yañez-Arenas et al., 2014*).

The utility of a species distribution model is dependent on the underlying species occurrence data used in constructing the model. Gaps and incomplete data can lead to misidentifying the target species' niche (*Monsarrat et al., 2019*). Underestimating species distributions can further mask the impacts of human activity on distributions, contributing to shifting baseline syndrome where progressively eroded species distributions or populations are accepted as normal or healthy (*Cromsigt, Kerley & Kowalczyk, 2012*). Ways to mitigate data gaps need thorough investigation.

Technological advances, global survey effort and digitisation of museum records have developed large biodiversity databases, pulling together disparate data sources to make global occurrence records more accessible and comprehensive. However, considerable gaps in biodiversity databases exist because of: detection difficulties, inconsistent surveying, and inadequate (or sometimes inaccurate) locality data for museum specimens (*Yesson et al., 2007*; *Beck et al., 2013*; *Troudet et al., 2017*).

Novel supplementary data sources could help fill biodiversity database gaps (*Toivonen et al., 2019*). With the proliferation of GPS enabled devices, the public is generating huge datasets on Web 2.0 platforms that can describe and/or predict a variety of phenomena including: protests (*Alanyali, Preis & Moat, 2016*), land-use (*Antoniou et al., 2016*), tourism (*García-Palomares, Gutiérrez & Mínguez, 2015*; *Chua et al., 2016*), hurricane damage (*Preis et al., 2013*) and protected area use (*Orsi & Geneletti, 2013*; *Hausmann et al., 2018*). Some Web 2.0 data, in the form of geo-tagged images, can communicate the identity and location of species (*Barve, 2014*). The geo-tagged images in a searchable social media platform may be a source for incidentally collected records of species (*Allain, 2019*; *Barve, 2014*; *ElQadi et al., 2017*; *Jiménez-Valverde et al., 2019*).

Incidental biodiversity records have already improved data availability for butterflies, snowy owls (*Barve, 2014*), spiders (*Jiménez-Valverde et al., 2019*), bees, flowers (*ElQadi et al., 2017*) and turtles (*Allain, 2019*). However, we do not know whether social media can provide similar benefits for (mostly) unpopular and low-detectability species.

Snakes present a model to explore the utility of social media data. Because snakes have been historically overlooked in research (*Shine & Bonnet, 2000*; *De Miranda, 2017*) and are difficult to detect (*Steen, 2010*; *Durso & Seigel, 2015*) they are likely suffering from a lack of primary biodiversity data. The need to generate more primary biodiversity data is underscored by: snakes' important regulatory and keystone roles in ecosystem functioning (*Willson & Winne, 2016*; *De Miranda, 2017*); major gaps in reptile conservation assessments (*Bland & Böhm, 2016*; *Tingley, Meiri & Chapple, 2016*; *Hughes, 2017*); and frequent involvement in human-wildlife conflicts (*Whitaker & Shine, 2000*; *Akani et al., 2002*; *Meek, 2012*; *Miranda, Ribeiro & Strüssmann, 2016*; *Marshall et al., 2018*).

We describe the current state of snake occurrence records in the Global Biodiversity Information Facility (GBIF) database, highlighting gaps in surveying. We then explore the potential utility of a supplementary data source, a photography sharing platform Flickr, in the modelling of tropical snake distributions.

## MATERIALS & METHODS

We completed all data analyses in R v.3.5.3 (*R Core Team, 2019*) and R Studio v.1.2.1335 (*R Studio Team, 2019*). For data visualisation we used ggplot2 (*Wickham, 2016*), ggrepel (*Slowikowski, 2019*) and scico (*Pedersen & Crameri, 2018*) packages. We have included a directory of scripts, packages (using packrat *Ushey et al., 2018*), and data used in analyses at: doi: 10.5281/zenodo.3243983.

### Data retrieval
#### GBIF

Before acquiring data from the GBIF database (*GBIF.org, 2019*) we generated a comprehensive list of snake species. We used the taxize package (*Chamberlain et al., 2018*) to access GBIF and National Center for Biotechnology Information (NCBI; *Benson et al., 2008*; *Sayers et al., 2009*) records for all squamates families, filtering for those mentioning Serpentes in downstream classification. For each family within the suborder Serpentes we then queried the GBIF database and downloaded occurrence records, on a per genus basis, using the dismo (*Hijmans et al., 2017*) and rgbif packages (*Chamberlain et al., 2019*). Once we had downloaded all records, we queried the GBIF database a second time to ensure that downloaded files were complete and included all available occurrences. All GBIF downloads and metadata (including a list of data sources) are available at doi: 10.5281/zenodo.3243983.

After downloading, we compiled the resulting genus occurrence files, filtering out marine snake families (data manipulation performed with the dplyr (*Wickham et al., 2017*), data.table (*Dowle & Srinivasan, 2019*) and reshape2 (*Wickham, 2007*) packages). Due to the size of the dataset, we automated cleaning. We opted to use the CoordinateCleaner package to clean each species individually (*Zizka et al., 2019*). Following the process outlined in *Zizka et al. (2019)*, we removed records with locations as NAs, zeros, identical values, near GBIF headquarters, near biodiversity institutions, within oceans, and that were extreme outliers for that species (using interquartile range outlier detection). Species with over 15,000 records (e.g., *Thamnophis spp.*, *Natrix spp.* and *Vipera berus*) failed or produced erroneous results so we examined these species manually, removing outlying occurrences (those occurring on incorrect continents).

#### Flickr

To acquire data from Flickr we generated a list of species names, both common and binomial as search terms. We used the taxize package (*Chamberlain et al., 2018*) to query the GBIF and NCBI databases for all species downstream of the Serpentes families. We then compiled results from both databases into a single list, removing duplicates. Common name queries of GBIF and NCBI databases were inadequate or failed for many species.

Therefore, we created a query system that accessed The Reptile Database (*Uetz, Freed & Hošek, 2019*) (using XML (*Lang & CRAN Team, 2019a*), xml2 (*Wickham, Hester & Ooms, 2018*) and rvest (*Wickham, 2019*)). For each species we retrieved all common names the Reptile Database listed. We parsed each set of common names to separate them to generate a list of search terms for each species, attempting to anticipate as many notation styles as possible used by Reptile Database. We relied on the stringr package (*Wickham, 2018*) to handle recurring character patterns.

We then accessed Flickr's API (*Flickr Development Team, 2019*), via R packages XML (*Lang & CRAN Team, 2019a*), RCurl (*Lang & CRAN Team, 2019b*) and httr (*Wickham, 2017*), using each species' search terms to retrieve search results for images. Photos had to be tagged as *snake* and geo-tagged so that a location was evident. During this process we saved the URL and year (extracted with the lubridate package *Grolemund & Wickham, 2011*) for each photo to later manually verify species identification. We then manually reviewed each image for the 18 selected tropical species (1166 images total), removing records of non-target species or images judged to have been taken within captive settings (captive settings were inferred by the presence of artificial substrate, white-balances associated with artificial lighting and geographic proximity to zoos, i.e., occurred within a raster cell and a suburban/urban area contiguous with a Google Maps labelled zoo).

### Raster layers

We used the raster package (*Hijmans, 2019*) to retrieve climatic raster data from WorldClim (*Fick & Hijmans, 2017*). To guard against over-parameterisation and over-fitting during species distribution modelling (*Fourcade, Besnard & Secondi, 2018*), we discarded 14 of WorldClim's bioclimatic layers. We discarded layers until between-layer correlations with an R value >0.6 were removed (*Merow, Smith & Silander, 2013*; *Castellanos et al., 2019*). We explored different combinations that reduced the correlation, and opted for a set of five variables covering a variety of climatic aspects likely important to snake range delimitation (*Kearney, Shine & Porter, 2009*; *Fourcade, Besnard & Secondi, 2018*). The remaining layers were: BIO1 (annual mean temperature), BIO2 (mean diurnal temperature range), BIO7 (temperature annual range), BIO12 (annual precipitation), and BIO15 (precipitation seasonality).

We limited the remaining WorldClim data to three regions of interest: Tropical Asia (longitude: 50°E, 150°E; latitude: −25°N, 50°N), Africa (longitude: −40°E, 40°E; latitude: −25°N, 75°N) and South America (longitude: −120°E, −25°E; latitude: −60°N, 25°N). We also downloaded global elevation data (*Danielson & Gesch, 2011*; *U.S. Geological Survey, 2016*) and human footprint index (*Venter et al., 2016a*; *Venter et al., 2016b*). Then we downscaled and reprojected elevation and footprint layers using projectRaster (*Hijmans, 2019*) with the default bilinear method to match the regional WorldClim layers' projection, extent and resolution. We have included the resulting raster layers used in analysis in the supplementary data.

## POINT PROCESS ANALYSIS

We examined the distribution of GBIF occurrences via several point process analyses. We set the data within a landmass polygon (to account for water bodies during calculations) downloaded from natural earth using the rnaturalearth package (*South, 2017*). Then using the spatstat package (*Baddeley & Turner, 2005*) we tested for spatial conformity. We performed four types of spatial tests. We ran quadrat tests (with quadrats roughly equivalent to 10 degrees squared) to examine the spatial randomness of points. We calculated nearest neighbour distance functions (G) with Kaplan–Meier, border, and hazard corrections to examine the distribution of distances from points to their nearest neighbour. We estimated the empty space function (F) with Kaplan–Meier, border, and Chiu-Stoyan correction to examine how empty space was distributed between points. Finally, we estimated Ripley's reduced second moment function (K), with no correction applied due to the prohibitively large dataset, for further examination of spatial non-randomness.

We estimated continental area and occurrence density using the rnaturalearth landmass. To estimate continental area, we projected the landmass for each continent using the closest Albers equal area conic projection (specifications obtained from https://epsg.io) with the rgeos (*Bivand & Rundel, 2018*) and sp packages (*Pebesma et al., 2019*). For standard error calculations we used the pracma package (*Borchers, 2019*).

## MODELLING

### Species selection

We selected 18 species to investigate: nine selected manually and nine selected randomly. Our manual selection was based on relative taxonomic stability, their charismatic appearance and ease of photographic identification: *Bitis arietans* MERREM, 1820; *Bothriechis schlegelii* (BERTHOLD, 1846); *Bungarus fasciatus* (SCHNEIDER, 1801); *Calloselasma rhodostoma* (KUHL, 1824); *Coelognathus radiatus* (BOIE, 1827); *Dendroaspis polylepis* GÜNTHER, 1864; *Eunectes murinus* (LINNAEUS, 1758); *Malayopython reticulatus* (SCHNEIDER, 1801); and *Ophiophagus hannah* (CANTOR, 1836). Our manual selection represented all three tropical regions (Tropical Asia: 5, Africa: 2, South America: 2).

In addition to the nine manually selected species, we used the sample_n function in dplyr (*Wickham et al., 2017*) to randomly select nine more species that fitted the following criteria: occurs entirely within one of the three tropical regions, and be considered taxonomically stable. We defined the second criteria using the names listed on Reptile Database. Any species with a single binomial name listed since 2000, we considered stable. Once we had filtered the list of species by those criteria, we randomly selected nine species from 25 species with the most Flickr results. We had to repeat the random selection to avoid species with too few occurrences to model or an insufficiently sized distribution to be estimated with the resolution of raster layers. *Porthidium spp* . also had to be excluded because of the difficulties verifying species identity in images. The final nine randomly selected species were: *Aplopeltura boa* (BOIE, 1828); *Atheris nitschei* TORNIER, 1902; *Boiga cynodon* (BOIE, 1827); *Boiga kraepelini* STEJNEGER, 1902; *Chironius carinatus* (LINNAEUS, 1758); *Echis coloratus* GÜNTHER, 1878; *Enhydris enhydris* (SCHNEIDER, 1799); *Hydrodynastes gigas*

(DUMÉRIL, BIBRON & DUMÉRIL, 1854); and *Sinonatrix percarinata* (BOULENGER, 1899) (Tropical Asia: 5, Africa: 2, South America: 2).

## Model settings

We created four training datasets per species. First, we used SPthin (*Aiello-Lammens et al., 2014*) with a grid size equal to the raster cell to thin the data, ensuring only a single occurrence per cell. We split the GBIF occurrences into five randomly assigned groups in geographic space, limiting non-independence in environmental space (*Roberts et al., 2017*; *Castellanos et al., 2019*). We used the BlockCV package (*Valavi et al., 2018*) with the recommended block size based on the climatic and elevation raster layers (using 100,000 samples, group assignment was optimised across 500 iterations). Where the recommended block size failed to assign at least one occurrence to every group, we decreased the block size by 5% and re-ran the assignment until all groups were represented. Once groups had been successfully assigned, we set aside the median sized group of points from testing. We used the remaining points to train the geo-independent model. We generated a second GBIF data-only training set with a random subset of the original data removed. We removed this subset with no space weighting (to replicate random k-folds frequently used in the modelling literature), and the size was equal to the subset removed for the geo-independent model training dataset. We refer to the second model as the GBIF random model. The final models used the two GBIF training datasets described above supplemented by the Flickr data collected for that species.

We generated an array of 10,000 background points for each species, the array was consistent between model runs and training datasets. We bounded background point generation with a minimum convex polygon around all species occurrence records (*Castellanos et al., 2019*), plus a buffer equal to half the mean distance between occurrences. Whereas studies usually choose a fixed buffer to create the bounding area, the disparity in our 18 species distributions required us to use species-specific buffers based on relative occurrence record spread. Relying on a compromised fixed buffer for all species could underestimate AUC scores for species with large distributions, while inflating AUC scores for species with small distributions (*Anderson & Raza, 2010*). Because survey effort is undocumented and unequal (*Tulloch et al., 2013*), we weighted background point distribution using a bias layer to areas that are likely to have had increased survey effort (*Phillips et al., 2009*; *Merow, Smith & Silander, 2013*). We chose human footprint as proxy for survey likelihood, under the assumption that increased access and human presence leads to greater occurrence records.

We used the ENMeval package (*Muscarella et al., 2014*) to run Maxent models across varying model settings. We chose Maxent because of its flexibility and performance relative to other methods (*Elith et al., 2006*). We used combinations of linear and quadratic feature classes and ran models using a sequence of regularization values from 1 to 8 to reduce the chances of overfitting (*Shcheglovitova & Anderson, 2013*; *Merow, Smith & Silander, 2013*; *Radosavljevic & Anderson, 2014*) and set internal cross validation to user groups defined with BlockCV (*Valavi et al., 2018*).

## Model evaluation

The metrics used to assess species distribution model performance are debated. Due to their reliance on pseudo-absences some of the ways of evaluating models are unhelpful. We chose to follow Castellanos et al.'s, (2019) advice and use multiple metrics. We selected receiver operating characteristic AUC (ROC AUC) because of its wide use and ability to compare models based on different datasets. Use of ROC AUC has drawbacks (*Lobo, Jiménez-valverde & Real, 2008*): it is sensitive to background area (*Anderson & Raza, 2010*), and is liable to overestimate model performance (*Fernandes, Scherrer & Guisan, 2018*). To supplement ROC AUC evaluation, we use precision-recall values (PRRC)—recently recommended as a metric insensitive to background area and species rarity because it ignores correctly predicted absences (*Sofaer, Hoeting & Jarnevich, 2019*). For every model created by the four training datasets, we calculated ROC AUC and PRRC values for all three test datasets with the PRROC package (*Grau, Grosse & Keilwagen, 2015*).

As an additional measure of the Flickr data's contribution to models, we examined the niche overlap between models trained on only GBIF records and those trained on datasets supplemented with Flickr occurrences. We estimated niche overlap using Schoener's D measure with the ENMeval package (*Muscarella et al., 2014*).

We explored Maxent model performance using GLM and GLMMs with the lme4 package (*Bates et al., 2015*). We created models using combinations of number of occurrences, species, and training dataset as predictors of PRRC and ROC AUC values. The full list of models tested can be found in Table S1. We used Spearman's rank test to explore the relationships between area and occurrence count, after testing for normality with qqplots (from the car package *Fox & Weisberg, 2011*) and Shapiro–Wilk tests.

## RESULTS

### Data summary

Our assessments of GBIF snake occurrences reveal strong spatial bias in the 302,386 unique locations of non-marine snakes. Flickr data searches produced only 44,689 images tagged with snakes and location information; Flickr data was also spatially nonuniform.

All point process analysis showed that the distribution of GBIF and Flickr points are not randomly distributed: multiple metrics suggest spatial clustering (GBIF data Quadrat test: $X^2 = 2425600$, $df = 288$, $p$-value $<2.2e-16$; Flickr: Quadrat test: $X^2 = 426820$, $df = 288$, $p$-value $<2.2e-16$; G-function: Fig. S1; F-function: Fig. S2; K-function: Fig. S3). The clustering is apparent in Figs. 1 and 2, illustrating points concentrated in North America, Europe and Australia –both GBIF and Flickr appear to follow similar distributions.

Examining the GBIF results per continent reveals the scale of spatial bias (Table 1). The number of occurrence records are considerably lower in Africa, Asia, and South America, despite their large area and diversity of snake species. This pattern is particularly apparent in the density of occurrence records that are approximately ten-fold lower.

The data available for our 18 selected snake species varied dramatically (Fig. 3), and appeared to only be weakly positively associated with the size of the minimum convex polygon (MCP) of occurrence points (Fig. 4). Overall, we manually reviewed 1166 Flickr

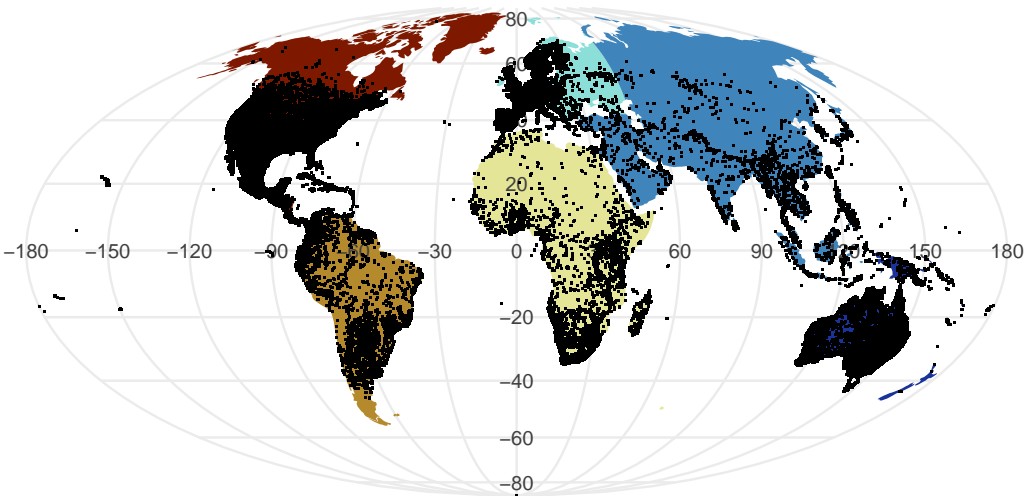

**Figure 1** Global distribution of all GBIF non-marine snake records displayed against continental divisions.

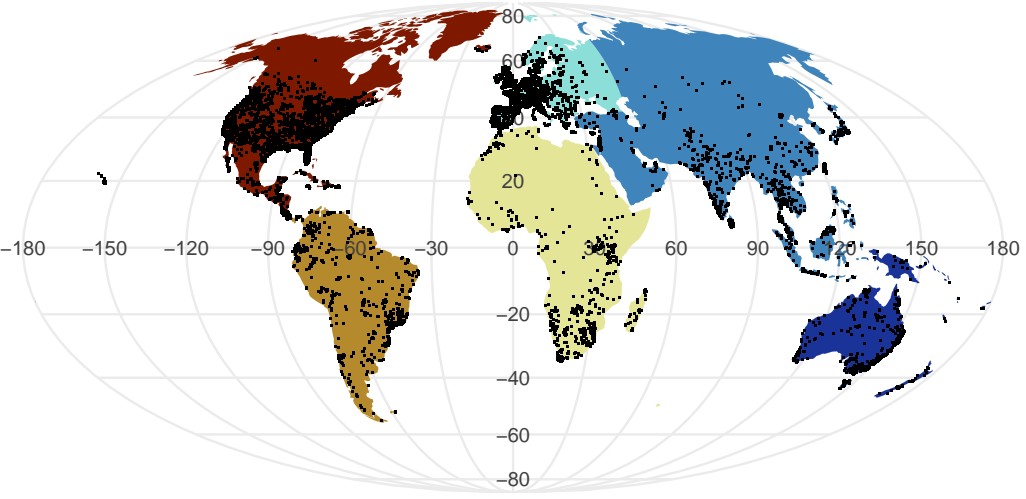

**Figure 2** Global distribution of geo-tagged Flickr photos that appeared across all searches.

images, discarding $11.22 \pm 5.68$ non-snake or captive image locations per species (range = 0–92; percentage of images discarded per species $11.06 \pm 4.86\%$, range = 0–77.97%).

## Modelling results

Overall, we found that models trained on GBIF supplemented with Flickr results were marginally better at predicting both randomly selected and geographically selected GBIF records when assessed using ROC AUC (Fig. 5). Precision-Recall values only saw the Flickr supplemented models perform better when predicting the geographically independent sample of GBIF records (Fig. 6).

**Table 1  Summary information of GBIF snake records.**

| Continent | # Species | # Occurrences | Mean occurrences per species | SE | Area (million km²) | Occurrences per 100 km² |
|---|---|---|---|---|---|---|
| Africa | 513 | 17,108 | 33.35 | 3.38 | 29.89 | 0.006 |
| Asia | 576 | 19,187 | 33.31 | 5.69 | 44.67 | 0.004 |
| Europe | 99 | 42,892 | 433.25 | 169.78 | 8.97 | 0.048 |
| North America | 680 | 157,923 | 232.24 | 33.62 | 24.64 | 0.064 |
| Oceania | 236 | 49,247 | 208.67 | 32.56 | 8.92 | 0.055 |
| South America | 633 | 16,029 | 25.32 | 2.21 | 17.91 | 0.009 |

**Notes.**
  # Species,  Number of species appearing in GBIF data, not the actual number of species known to exist across the continent; # Occurrences,  Number of occurrence records in GBIF downloads; Mean occurrences per species,  Total number of occurrences records in a continent divided by number of species in GBIF data; SE,  Standard error associated with the mean occurrences per species;  Area,  Area in millions of km² estimated using Albers equal area conic projection; Occurrences per 100 km²,  Total number of occurrence records divided by the estimated continental area.

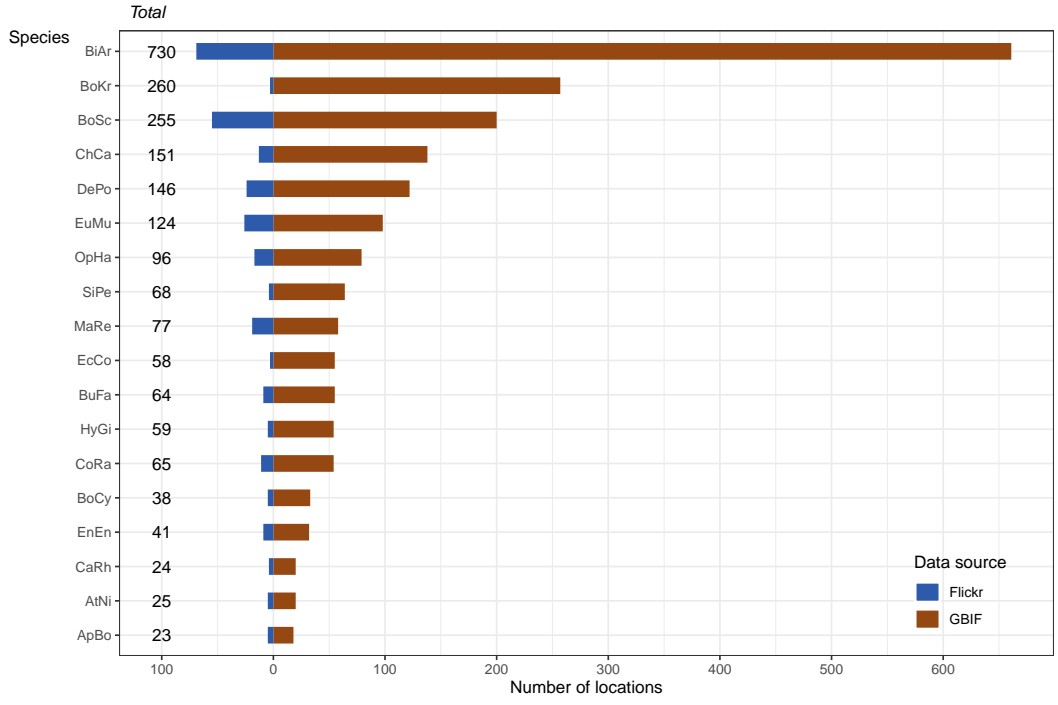

**Figure 3  Number and source of selected species occurrence records.** Species codes in order of appearance: BiAr = *Bitis arietans*, BoKr = *Boiga kraepelini*, BoSc = *Bothriechis schlegelii*, ChCa = *Chironius carinatus*, DePo = *Dendroaspis polylepis*, EuMu = *Eunectes murinus*, OpHa = *Ophiophagus hannah*, SiPe = *Sinonatrix percarinata*, MaRe = *Malayopython reticulatus*, EcCo = *Echis coloratus*, BuFa = *Bungarus fasciatus*, HyGi = *Hydrodynastes gigas*, CoRa = *Coelognathus radiatus*, BoCy = *Boiga cynodon*, EnEn = Enhydris enhydris, CaRh = *Calloselasma rhodostoma*, AtNi = *Atheris nitschei*, ApBo = *Aplopeltura boa*.

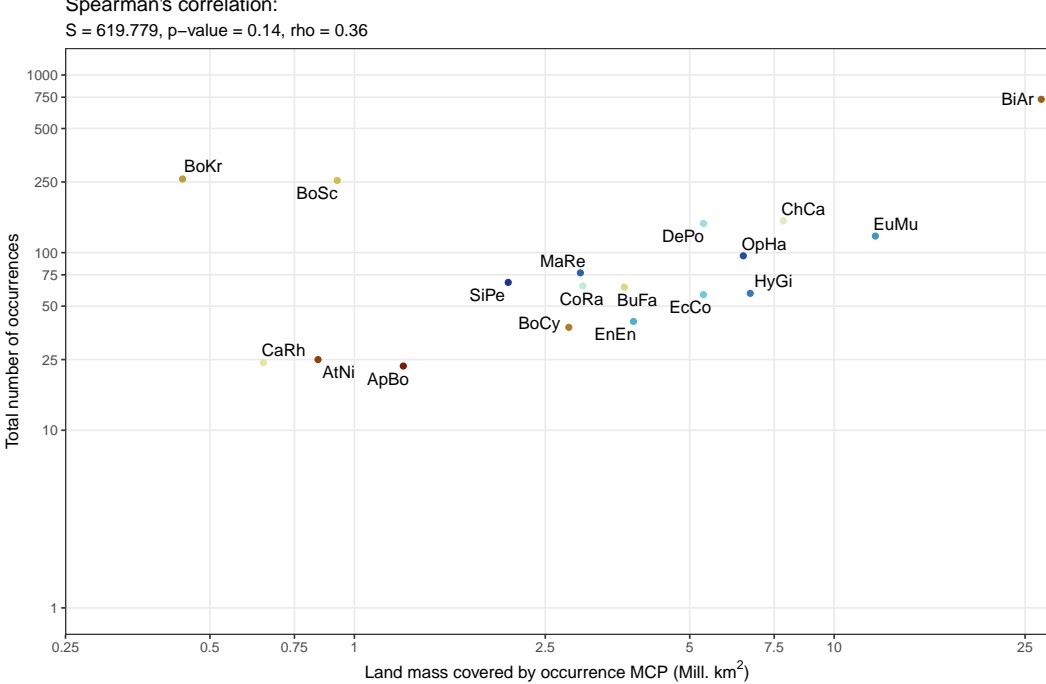

**Figure 4  Relationship between number of occurrences and the minimum convex polygon (MCP) area cover by occurrence points.** Minimum convex polygon are clipped to exclude oceans. Both scales are presented as logs to make individual species visible. Species codes from left to right: BoKr = *Boiga kraepelini*, CaRh = *Calloselasma rhodostoma*, AtNi = *Atheris nitschei*, BoSc = *Bothriechis schlegelii*, ApBo = *Aplopeltura boa*, SiPe = *Sinonatrix percarinata*, BoCy = *Boiga cynodon*, MaRe = *Malayopython reticulatus*, CoRa = *Coelognathus radiatus*, BuFa = *Bungarus fasciatus*, EnEn = *Enhydris enhydris*, DePo = *Dendroaspis polylepis*, EcCo = *Echis coloratus*, OpHa = *Ophiophagus hannah*, HyGi = *Hydrodynastes gigas*, ChCa = *Chironius carinatus*, EuMu = *Eunectes murinus*, BiAr = *Bitis arietans*.

Models trained on randomly and geographically independent GBIF data performed similarly when tested against the Flickr data. The randomly subset GBIF models showed more variable results both for ROC AUC and PRRC. The respectable ability to predict Flickr results from only GBIF records suggest that Flickr results have little in the way of new climatic information.

The limited new information provided by Flickr datasets is further supported by the high levels of niche overlap between models trained on GBIF-only and Flickr-supplemented datasets, albeit with variation between species (Fig. 7).

When we investigated which variable predicts model performance, the mixed-models using the training dataset and species as random predictors were superior based on AIC. The resulting model agrees with Figs. 1 and 2 indicating variability between species and a weak trend driven by the training dataset. While the model investigations seem to support species as the driver behind Maxent model performance, the residuals from the models remain highly structured and non-normal (Sharpiro-Wilk test: PRRC as response, $W = 0.7526$, $p$-value $< 2.2e-16$; ROC AUC as response, $W = 0.94194$, $p$-value $< 2.2e-16$). Our models exploring change in model evaluation metrics suggested that the difference in sample

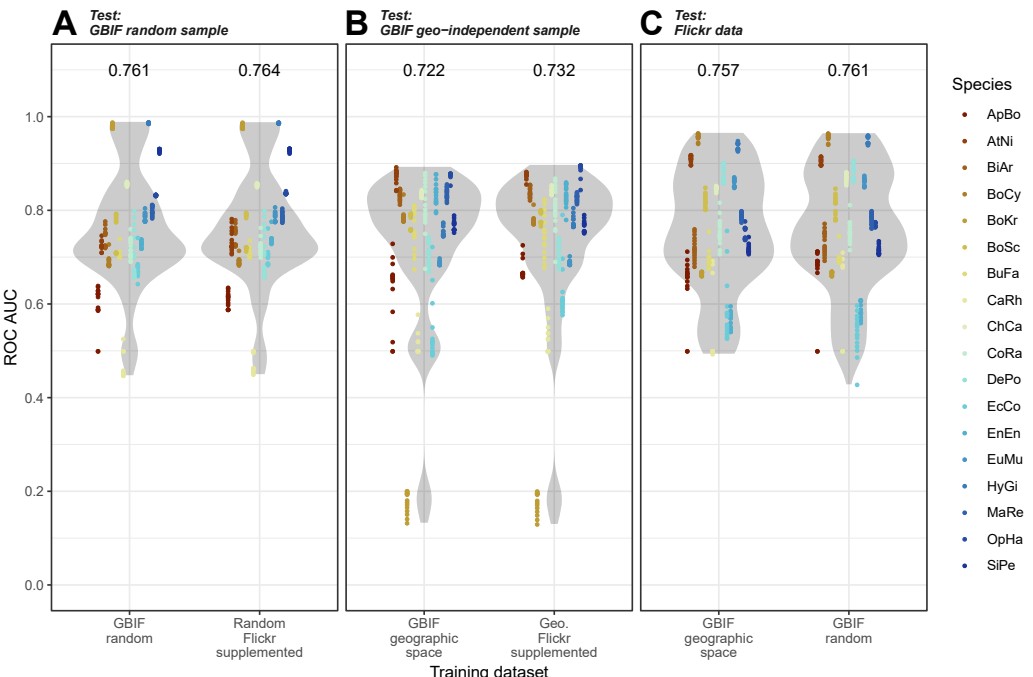

**Figure 5** **Receiver Operating Characteristic results for the three models when tested against the three independent test datasets.** (A) GBIF random sample. (B) GBIF geo-independent sample. (C) Flickr data. Species codes in alphabetical order: ApBo = *Aplopeltura boa*, AtNi = *Atheris nitschei*, BiAr = *Bitis arietans*, BoCy = *Boiga cynodon*, BoKr = *Boiga kraepelini*, BoSc = *Bothriechis schlegelii*, BuFa = *Bungarus fasciatus*, CaRh = *Calloselasma rhodostoma*, ChCa = *Chironius carinatus*, CoRa = *Coelognathus radiatus*, DePo = *Dendroaspis polylepis*, EcCo = *Echis coloratus*, EnEn = *Enhydris enhydris*, EuMu = *Eunectes murinus*, HyGi = *Hydrodynastes gigas*, MaRe = *Malayopython reticulatus*, OpHa = *Ophiophagus hannah*, SiPe = *Sinonatrix percarinata*.

size played a very small role (negative relationship with PRRC values: $-0.0095 \pm 0.0039$, $p = 0.015$; positive relationship with ROC AUC values: $0.0260 \pm 0.0042$, $p < 0.001$) and the changes were largely dependent on the species (model specification and AIC values can be found in Table S1).

## DISCUSSION

### Spatial bias

Our results show a strong spatial bias in GBIF's occurrence records for non-marine snakes. The lack of records in the critical snake hotspots mirrors investigations into other taxonomic groups (*Yesson et al., 2007*; *Amano, Lamming & Sutherland, 2016*; *Roll et al., 2017*). The identified gaps in GBIF records support efforts to make use of more diverse data sources: by filling gaps in GBIF coverage and boosting sample sizes, supplementary data sources could reduce the chances of underestimating species distributions and ecological niches (*Beck et al., 2013*; *Monsarrat et al., 2019*). However, while our efforts to retrieve occurrence records from social media were successful, the quantity of records was insufficient to make significant impacts on distribution models. The gaps in GBIF records

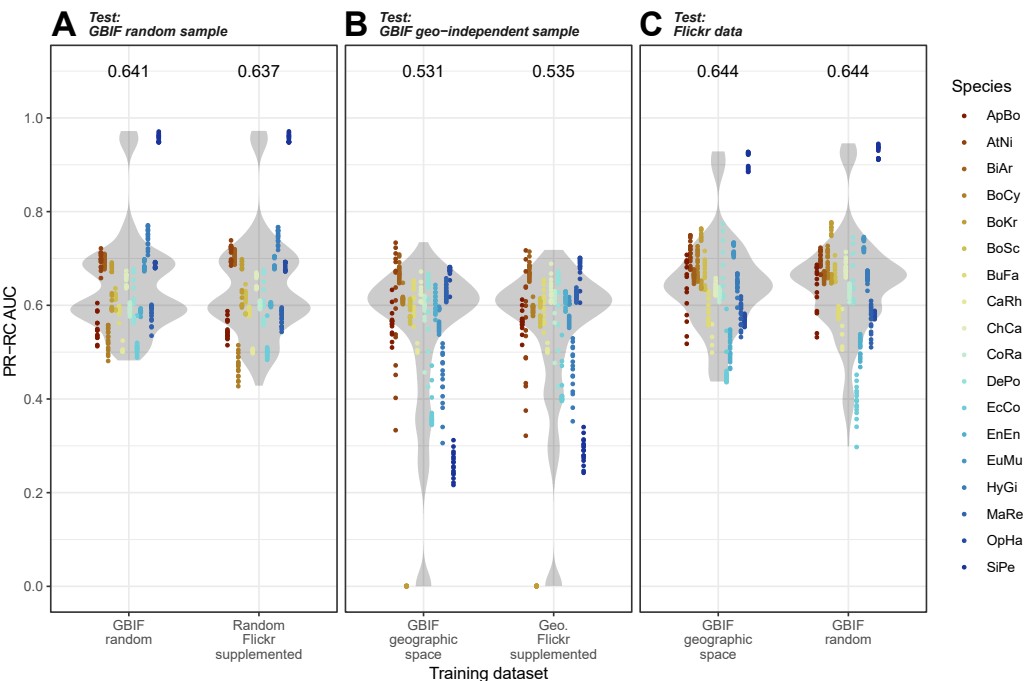

**Figure 6** **Precision-Recall results for the three models when tested against the three independent test datasets.** (A) GBIF random sample. (B) GBIF geo-independent sample. (C) Flickr data. Species codes in alphabetical order: ApBo = *Aplopeltura boa*, AtNi = *Atheris nitschei*, BiAr = *Bitis arietans*, BoCy = *Boiga cynodon*, BoKr = *Boiga kraepelini*, BoSc = *Bothriechis schlegelii*, BuFa = *Bungarus fasciatus*, CaRh = *Calloselasma rhodostoma*, ChCa = *Chironius carinatus*, CoRa = *Coelognathus radiatus*, DePo = *Dendroaspis polylepis*, EcCo = *Echis coloratus*, EnEn = *Enhydris enhydris*, EuMu = *Eunectes murinus*, HyGi = *Hydrodynastes gigas*, MaRe = *Malayopython reticulatus*, OpHa = *Ophiophagus hannah*, SiPe = *Sinonatrix percarinata*.

(and similar gaps in Flickr derived data) are likely not the results of lack of knowledge in these locations (*Tantipisanuh & Gale, 2018*), but barriers limiting submissions to global biodiversity databases (*Amano & Sutherland, 2013*).

Other studies had highlighted the potential of social media photographs to supplement existing occurrence records (*Allain, 2019*; *Barve, 2014*; *ElQadi et al., 2017*), but stopped short of exploring how the records would impact distribution modelling and model predictive power. Studies that explored the impact on models' predictive power targeted more readily photographed species in a region with greater interaction with biodiversity recording (*Jiménez-Valverde et al., 2019*). Tropical snakes provide a harsher assessment of the utility of community generated geo-tagged images. Our findings suggest that while there is a growing potential for social media to supplement biodiversity databases, the benefits are currently minimal for species with low-detectability and vary dramatically between species.

There are several reasons for researchers to consider using social media despite the marginal impacts shown here. First, is the low cost of initially screening for potential records. Flickr's map user interface (https://www.flickr.com/map) can be used to gauge the

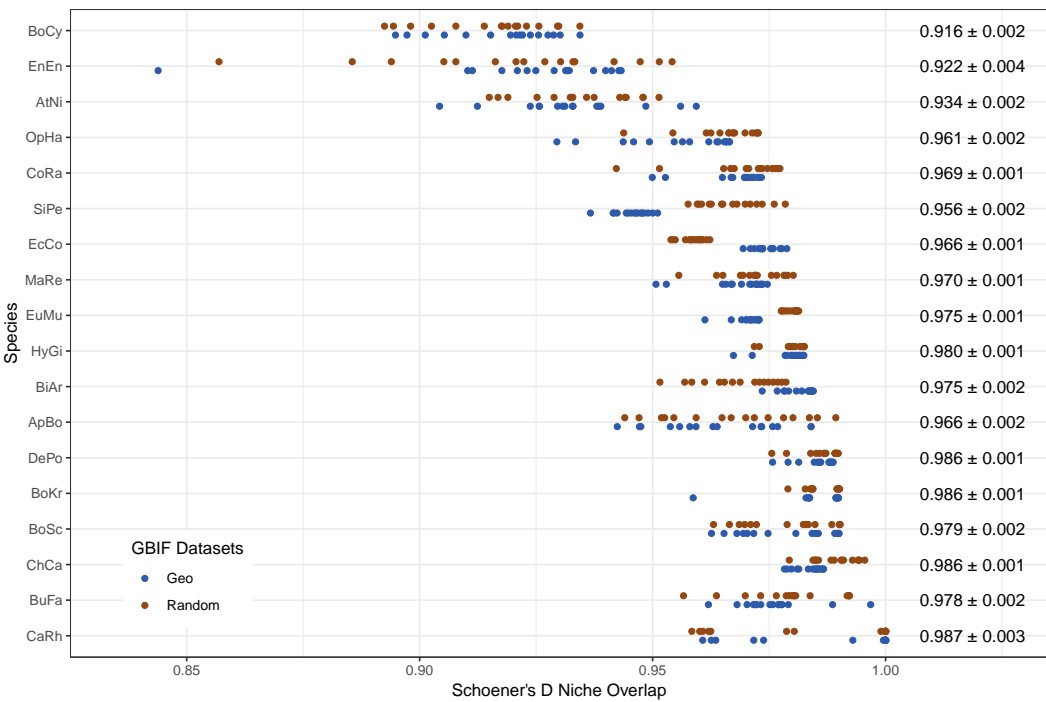

**Figure 7** **Schoener's D measure of niche overlap for models trained on GBIF-only and Flickr supplemented datasets.** Right-hand side values show the overall niche overlap mean per species and the standard error. Species codes top to bottom: BoCy = *Boiga cynodon*, EnEn = *Enhydris enhydris*, AtNi = *Atheris nitschei*, OpHa = *Ophiophagus hannah*, CoRa = *Coelognathus radiatus*, SiPe = *Sinonatrix percarinata*, EcCo = *Echis coloratus*, MaRe = *Malayopython reticulatus*, EuMu = *Eunectes murinus*, HyGi = *Hydrodynastes gigas*, BiAr = *Bitis arietans*, ApBo = *Aplopeltura boa*, DePo = *Dendroaspis polylepis*, BoKr = *Boiga kraepelini*, BoSc = *Bothriechis schlegelii*, ChCa = *Chironius carinatus*, BuFa = *Bungarus fasciatus*, CaRh = *Calloselasma rhodostoma*.

number of potential records before undertaking the task of extracting (and reviewing) the records. Second, are the benefits of increased sample sizes that analyses of real-world data find difficult to quantify. Larger samples are less sensitive to false-positives/negatives and locational error (*Wisz et al., 2008*; *Mitchell, Monk & Laurenson, 2017*; *Fernandes, Scherrer & Guisan, 2018*). When working with species with fewer than 20-30 records, model performance is more likely to be improved by any additional records (*Stockwell & Peterson, 2002*); only three of our tested species had fewer than 30 records. Third, species distribution modelling techniques can vary in their sensitivity to changes in sample size (*Thibaud et al., 2014*; *Fernandes, Scherrer & Guisan, 2018*); Maxent tends to be a less sensitive technique (*Thibaud et al., 2014*).

## Supplementary data sources limitations and potential

We highlight three limitations to implementing social media occurrence into species distribution efforts.

First is the number of geo-tagged images for low detectability species. The species with the most photographs relative to GBIF records tended to be more striking, either in size or colouration (e.g., *Eunectes murinus, Malayopython reticulatus* and *Bothriechis*

*schlegelii*); a pattern reflected in GBIF records overall (*Troudet et al., 2017*). Public interest in reptiles has also been linked to whether a species is venomous, endangered, or widely distributed (*Roll et al., 2016*). It may be the case that traits associated with people's interest in a species are mediated by traits that control how likely a species is to be photographed, such as its rarity or natural history (e.g., generalist species may be more photographed than purely cryptozoic species). Limitations associated with the quantity of photos will lessen over time as GPS enabled cameras become more common and the growth in geo-tagged images continues to increase (Fig. S4). Accessing other social media platforms containing geo-tagged images could additionally bolster occurrence datasets. However, current terms and conditions on several potential platforms prohibit data mining or have significant barriers to data access (*Toivonen et al., 2019*). Reliance on manual curation of occurrence records may be feasible when focusing on a single species but will become prohibitively time-consuming when assessing a wider clade.

The second limitation is the need to verify the identity of species depicted. While community science projects can have good identification rates for non-professional participants (*Austen et al., 2016*; *Kosmala et al., 2016*), species distribution modelling can be sensitive to false-positives (*Fernandes, Scherrer & Guisan, 2018*). Eliminating false-positives currently requires manual verification by the researchers, but there is significant progress being made in automated species identification (*Botella et al., 2018*; *Wäldchen & Mäder, 2018*; *Toivonen et al., 2019*). For snakes, a reliable system may be difficult to perfect given their crypsis and current taxonomic fluidity. Even if automated photographic verification can become reasonably reliable, it would be prudent to explicitly integrate the confidence of species identification into the distribution models, a practice that has already been demonstrated to improve predictions (*Louvrier et al., 2018*; *Johnston et al., 2018*).

Finally, researchers must consider the drivers behind different data sources distributions (*Li, Goodchild & Xu, 2013*). The use of bias layers in presence only modelling is the primary way to mitigate the impacts of an unknowable survey effort (*Phillips et al., 2009*; *Merow, Smith & Silander, 2013*). However, bias layers derived from the spatial patterns of one dataset may be inappropriate for another. This is why we opted for a bias layer, human footprint, that is likely connected to the overall distribution of wildlife observations. With larger datasets from more sources there may be a need to account for sampling bias on a per-dataset basis. Alternatively, social media derived datasets could be used only in model validation, proving a "semi-independent" dataset to supplement cross-validation (*Gregr et al., 2019*).

## Conservation implications

Numerous reptiles lack proper conservation assessment due to data deficiency (*Bland & Böhm, 2016*). Discovering ways to fill data gaps (e.g., *Callaghan et al., 2019*) without having to fund additional surveying efforts would be valuable at a time when natural history investigations are under appreciated but macro-ecological questions are popular (*Ríos-Saldaña, Delibes-Mateos & Ferreira, 2018*; *McCallen et al., 2019*). Overcoming data deficiencies should be prioritised; delays could result in occurrence data derived from distributions defined by human activity (realised niche), rather than the climatic or

absolute niche of a species (*Monsarrat et al., 2019*). Improvements in occurrence data may help identify current distributions, but unstructured occurrence data cannot help quantify population trends sorely needed for many reptile species (*Bland & Böhm, 2016*; *Bayraktarov et al., 2019*).

The quantity and accessibility of social media species occurrence records is open for abuse. In herpetology, there have been several cases of species being negatively affected by the scientific publication of location data (*Stuart et al., 2006*; *Lindenmayer & Scheele, 2017*) even though journals allow masked or partial publication (*Lowe et al., 2017*). While there is understandable fear in publishing the locations of new and desirable species in scientific literature, how long does it take for that information to enter the public sphere via geo-tagged photography? With the rapid growth geo-tagged images, being able to keep a desirable species protected by secrecy or gate-keeping may become increasingly difficult.

## CONCLUSION

We have highlighted that there is considerable spatial bias in the GBIF records for non-marine snakes, with gaps in tropical regions that house exceptionally high snake diversity (*Roll et al., 2017*). While we encourage the investigation of supplementary data sources to help fill gaps in biodiversity databases, currently accessible social media occurrence records only improve species distribution models marginally. The data availability for tropical snakes is highly variable between species and emphasises the difficulties researchers face when studying low detectability species. Both GBIF and social media data sources are growing exponentially (although not uniformly across taxa *Amano, Lamming & Sutherland, 2016*); tapping the full potential of these resources may be best realised with integration of image recognition and identification confidence.

## ACKNOWLEDGEMENTS

We thank the Suranaree University of Technology for providing the resources required to undertake this research. We thank Inês Silva and Matt Crane for enduring long discussions on model evaluation metrics. We thank the Flickr team for creating an API that is accessible and searchable. We thank countless photographers across the globe for their enthusiasm for wildlife. We would also like to thank the editor, Assoc. Prof. Alastair Culham, and three anonymous reviewers for their comments and insight in improving this manuscript.

### Funding

This research was supported by the Suranaree University of Technology, Insitute of Research and Development. The funders had no role in study design, data collection and analysis, decision to publish, or preparation of the manuscript.

### Grant Disclosures

The following grant information was disclosed by the authors:
Suranaree University of Technology, Insitute of Research and Development.

## Competing Interests

The authors declare there are no competing interests.

## Author Contributions

- Benjamin M. Marshall conceived and designed the experiments, analyzed the data, prepared figures and/or tables, authored or reviewed drafts of the paper, approved the final draft.
- Colin T. Strine conceived and designed the experiments, authored or reviewed drafts of the paper, approved the final draft.

## Data Availability

We have uploaded all data and code used in this study to Zenodo at DOI: 10.5281/zenodo.3243983.

The code does not include API keys to access data services. A Flickr API key can be acquired from https://www.flickr.com/services/apps/create/ after creating a Flickr account and submitting a request. There is documentation on obtaining and using the API at https://www.flickr.com/services/api/. A NCBI API key can be acquired from the accounts page https://www.ncbi.nlm.nih.gov/account/settings/ after creating an account with the NCBI. The API keys are only required for data curation code segments, we have included the datasets we curated during the study to allow the analysis code segments to run independently.

## Supplemental Information

Supplemental information for this article can be found online at http://dx.doi.org/10.7717/peerj.8059#supplemental-information.

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
