# Peer review of "Exploring snake occurrence records: Spatial biases and marginal gains from accessible social media"

_PeerJ, doi:10.7717/peerj.8059_

## Round 0.1 · original submission · Major Revisions

This paper is publishable but not in its current form. All reviewers agreed there was interesting material in the manuscript but all felt there were areas for improvement. You need to pay particular attention to making sure the storyline of the paper flows better so that it will be easier to understand the later findings. You should probably simplify the tests you use and think carefully which is most appropriate to the data and question. You must also set your work in a braoder context of the various large and international CS data programmes. However I would very much welcome a revised manuscript and think there is great potential to publish in due course.

Reviewer 1 ·

Basic reporting

The manuscript is well written, clear and structured. The authors give a succinct summary of the relevant background information. I would be interested in a little more elaboration on which taxa social media records have been most successful in providing information on but this is a minor point - all the relevant citations are given in the text.

Experimental design

The methods are generally well described and code is provided.

Lines 173-181. I got a little confused here. It sounds like you randomly selected 9 species and then randomly selected a further 9. But I don’t think you did? Could you rewrite this paragraph to make it clear please? Thanks.

Validity of the findings

The results are given clearly and concisely.

Additional comments

Might there be other factors which explain the number of occurrence records? Rarity, threat status, body size, habitat use or other ecological traits? I appreciate you do say in the discussion that species that are brightly coloured/large have more Flikr records.
Line 61: You are missing a fullstop.
Line 83: Typo, ggplot2

Line 348: ‘data deficiency’ not ‘deficient data’

Reviewer 2 ·

Basic reporting

In general, I think the paper is well written, clear and unambiguous. The hypotheses are well presented. The list of references cited could be a bit more complete. I detected two English mistakes in line 95 “all recorded (records)” and line 355 “However, while improvements in occurrence data my (may) help identify current distributions…”
The last sentence of the abstract is not consistent with the results obtained “We suggest that researchers studying species with low detectability or limited data consider integrating social media records into occurrence datasets (if available), so long as photographic identification is viable.”

Experimental design

The research presented suits well the aims and scope of the journal. The research question is well defined, relevant and meaningful. This paper is anchored in an area of research in wide expansion and that relies on information collected by citizens in social media. However, I missed the inclusion of data from global citizen science projects such as iNaturalist, iSpot, Atlas of Living Australia, Observado among others. I believe that the results obtained could have been different if this information were also used. Opportunistic citizen science databases of species observations can represent a viable alternative to scientific records when these are not available, and the challenge might be to combine different data sources to achieve better results. Materials and Methods were described with sufficient detail allowing the results to be replicate. The way they were presented shows rigorous investigation.

Validity of the findings

With the experimental design presented I felt a bit disappointed with the results obtained. The paper “highlighted that there is significant spatial bias in the GBIF records for non-marine snakes, with gaps in tropical regions”. The authors encourage the investigation of supplementary data sources to help fill gaps in biodiversity databases, currently accessible social media occurrence records only improve species distribution models marginally.” I would like to see a bit more of this supplementary data sources studied like, for instance citizen science data or other social networks.

Additional comments

I like the paper, the question raised, and approach followed by the authors however, since the results do not seem very strong, I would like to see some more analysis performed with other data sources.

Reviewer 3 ·

Basic reporting

Currently, I don’t feel the introduction flows particularly well. To address this I suggest a broader opening paragraph in the introduction discussing the value that SDMs have in the ecological literature. Add some examples of the types of research they have been used for, this could include the current introduction section on threat assessments. The next section could be focussed on the data sources and issues surrounding them for SDMs. Next you could add the introduction of snakes as a model system here. Finally, end with a clear set of aims for the study.

Experimental design

Generally, the methods are clear and easy to follow. However, there are a couple of points that need additional detail to ensure readers can replicate the study.

Line 129: In the interest of reproducibility, how was “geographic proximity to zoos” assessed? For example, were photos within 1km of a zoo excluded?

Line 145: Please describe how these variables were downscaled.

Validity of the findings

The addition of the social media data resulted in only marginally different niche estimations and negligible difference in Maxent model performance, however the authors still recommend readers acquire the extra data sources. I feel an obvious question is why bother going through the effort to extract extra data sources when they appear to have little impact on the result? Particularly given the difficulty in accessing these data and ensuring their reliability (presumably this was not a quick job – Line 126: suggests manual verification of at least 45000 snake photos!). I feel a discussion around this question is important, and perhaps a line in the abstract summarising why we should seek additional data would be useful.

Additional comments

The authors present an interesting study where they assess the added value of including distribution data scrapped off social media (geo-tagged species images from Flickr in this case), to species distribution model performance. The key result was that SDMs which included data derived from social media had only a very minor improvement in model performance compared to those based purely on GBIF data. The authors rightly highlight the need for macroecological studies to utilise as many data sources as possible, particularly when studying species with low detectability. Please see my additional comments on the study below:

Firstly, the code to extract and clean data from GBIF is useful, however the code for extracting images from Flickr is exceptionally useful, particularly given the main message of this paper (i.e. use these other data sources!). I feel the authors could make more of this in the paper, for example add a sentence to the abstract saying “we provide code for extracting geo-tagged images of species from Flickr”. This code is not only relevant for the types of study we see here, but also to those interested in putting together photo libraries for training image recognition algorithms, a growing area in ecological research. I feel this simple change would add value to this study, increasing citations of the paper and the associated code.

Minor comments:
Lines 41-49: Do the IUCN actually accept SDM outputs as part of the species assessments?

Line 66-68: The two questions seem oddly placed here. Suggest rephrasing this from a question to a point, i.e. it is not known how…

Line 131: missing “(“ before Fick.

Line 262: Please add the direction to this association, i.e. weakly positively associated.

Line 355: typo- “may” not “my”

Figure 3: Could this be ordered by number of GBIF records. I feel this would emphasise the variation in ranking (based on n records) when compared to Flickr.

---

## Round 0.2 · accepted · Accept

Both the reviewers and I are fully satisfied with the changes you have made. Thank you.

Reviewer 1 ·

Basic reporting

I am happy with the changes the authors made in response to my comments.

Experimental design

I am happy with the changes the authors made in response to my comments.

Validity of the findings

I am happy with the changes the authors made in response to my comments.

Additional comments

I am happy with the changes the authors made in response to my comments.

Reviewer 2 ·

Basic reporting

I think the new version of the paper answers the questions raised and clarifies the results obtained. I think the publication is appropriate and interesting for the scope of the journal. The new literature cited is adequate and responds to the requests of the reviewers

Experimental design

The reviewers' suggestions were taken into consideration and the answers given were enlightening.

Validity of the findings

The major problems identified have been resolved.

Additional comments

I like the paper, the question raised, and approach followed by the authors. The authors resolved the questions raised by the reviewers